# Effects of Footwear Selection on Plantar Pressure and Neuromuscular Characteristics during Jump Rope Training

**DOI:** 10.3390/ijerph20031731

**Published:** 2023-01-18

**Authors:** Hai-Bin Yu, Wei-Hsun Tai, Ben-Xiang He, Jing Li, Rui Zhang, Wei-Ya Hao

**Affiliations:** 1Graduate School, Chengdu Sport University, Chengdu 610000, China; 2School of Physical Education, Quanzhou Normal University, Quanzhou 362000, China; 3College of Textiles and Apparel, Quanzhou Normal University, Quanzhou 362000, China; 4Key Laboratory of Bionic Engineering (Ministry of Education, China), Jilin University, Changchun 130022, China; 5China Institute of Sport Science, General Administration of Sport of China, Beijing 100061, China

**Keywords:** footwear, jump rope, plantar pressure, electromyography

## Abstract

This study examined what footwear type influenced plantar pressure and lower extremity muscle activations in jump rope training. Ten healthy physical-education graduate students participated in this study. The biomechanical parameters during the jump rope training were collected by an AMTI force platform, a Novel Pedar-X insole and a wireless electromyography (EMG) system. The results of the force platform indicate that vertical ground reaction force (vGRF) and contact time were much higher in the one-leg landing (both *p* = 0.001). The GRF, GRF (BW) and Lat MF pressure were significantly greater in the one-leg landing (*p* = 0.018, 0.013 and 0.027); the pressure of the Lat MF and H area were significantly greater in the volleyball shoe (*p* = 0.025, 0.031); the pressure of the Mid FF and Lat FF area were significantly greater in the jumping shoe (*p* = 0.005, 0.042). No significant difference in EMG was found between footwear and landing conditions. In summary, the running shoe and jumping shoe might be a better choice for people who exercise. However, the running shoe is recommended for people when both jumping and running are required.

## 1. Introduction

Good physical fitness is needed to carry out daily tasks and physical activities both now and in the future [1,2]. A report from the World Health Organization (WHO) indicated that 80% of the young population is lacking physical activity, and a third of the world’s population now has a sedentary lifestyle [3]. Regular physical training has numerous beneficial adaptations, such as disease prevention, which also decreases the risk of a sedentary lifestyle [4]. Furthermore, good physical fitness may offer a competitive advantage [5,6] and a reduced risk of injury [7,8]. However, physical training opportunities are always limited by external requirements such as equipment, time or space. Therefore, jump rope training (JRT) may provide a simple and accessible exercise, the idea of which is developed from primary schools [9].

Jump rope is a safe and high-efficient training method [10,11,12] that is also suitable for any health and subhealth populations regardless of sex and age. Furthermore, JRT also provides improvements in bone mineral content [13] and reactive strength and stiffness [11], the outcome of which can be seen with traditional plyometric training [14]. Previous research indicated that JRT could be more effective in larger groups because of its reduced activity space and low-budget equipment requirements [11]. However, it is important to avoid acute injury or chronic damage during JRT in order to gain the aforementioned benefits, and the choice of appropriate footwear is important to avoid stress injuries from repeated shock [15].

Footwear is essential to the human plantar. Shoes are a highly effective tool that solve the challenges of daily routine. Shoes can provide stability, flexibility or rigidity and comfort; however, in some situations, they can even cause injury to feet [16]. Therefore, a suitable choice of shoe for specific exercises is important and can also decrease the risk of injury [17]. In recent years, shoes have provided comfort, performance enhancement and a reduced occurrence of injury for those who wear them, such as the general population and athletes [18], yet most research has focused on running [19]. Less research has focused on actions such as rope jumps, and, just as with running, this exercise should also avoid continuous sub-maximal loading and force shocks, which can lead to musculoskeletal injury [20]. Therefore, it is important to understand what type of shoe is required during continuous jumping-related exercises.

Based on previous biomechanical research, it is suggested that footwear which reduces more plantar pressure will be better for long-term jump rope exercises [21]. To recommend what footwear people should use, the purpose of the present study was to focus on examining the effect of different footwear types on jump rope biomechanics and offer information regarding the selection of appropriate footwear for daily exercise.

## 2. Materials and Methods

### 2.1. Participants

There were 5 male and 5 female physical education graduate students who voluntarily participated in this research (age: 25.4 ± 3.9 years; body weight: 60.1 ± 6.4 kg and height: 1.66 ± 0.58 m). Participants were asked if they had the ability to achieve 140 single under-rope jumps in a min prior to testing. The G*Power (version 3.1.9.2; Franz Faul, Universität Kiel, Kiel, Germany) was used to calculate sample size with a power level of 71% and an alpha level of 0.05. All the participants had been free of any lower limb injuries for 6 months, and they were required to understand and sign an informed consent before the experiment. The full study procedures and potential risks were explained to them. The research was conducted in accordance with the Declaration of Helsinki, and the procedures were approved by the Institutional Review Board of University of Taipei (IRB-2016-021).

### 2.2. Instruments

An AMTI force platform (1000 Hz, BP600900-6-2000; Advance Mechanical Technology, Inc., Phoenix, AZ, USA) was used to collect the ground reaction force; a Novel Pedar-X insole (100 Hz; Nove GmbH, Munich, Germany) was used to collect plantar pressure, which was distributed through 99 force sensors contained in the insole and had a measuring range up to 600 kPa. The insole was divided into eight regions, which followed the recommendations of a previous study [21]. The muscle activations of the vastus medialis (VM), biceps femoris (BF), tibialis anterior (TA) and gastrocnemius muscle (GA) were collected by using a Noraxon wireless surface electromyography (EMG) system (1500 Hz; Noraxon USA Inc., Scottsdale, AZ, USA).

### 2.3. Procedures

There was a 10 min warm up to help accustom the participants to the insoles and the different types of shoes; after that, the shoes were randomly assigned. The EMG electrodes were on the dominant leg, the side that was used to kick a ball [22], and Pedar-X insoles were placed on bilaterally. Participants were then tested on maximal voluntary isometric contraction (MVC) for 5 s with a maximal voluntarily contracted implement that was used to calculate normalization with four muscles. Each participant was asked to perform a one- and two-leg jump rope randomly with three types of footwear (Li Ning-ARHH049-3 running shoe, Jingyuan-TS001 jumping shoe and Asics-river ex7 volleyball shoe) that in Figure 1. They were asked to jump under a 2.2 Hz frequency with a metronome guidance lasting for 30 s. Five consecutive cycles (touchdown—airtime—touchdown) of jump rope exercises were recorded for analysis from each trial. A rest period of >3 min between each shoe condition.

### 2.4. Data Processing

Touchdown and airtime were measured by the force platform, and the threshold of vertical ground reaction force was set at 10 N. The Pedar Multiproject-ip software (Novel Electronics, St. Paul, MN, USA) was used to analyze plantar force, plantar pressure, pressure–time integral and force–time integral which divided the foot into 8 regions for measurement. These regions were: hallux (H), lesser toes (LT), medial forefoot (Med FF), middle forefoot (Mid FF), lateral forefoot (LatFF), medial midfoot (Med MF), lateral midfoot (Lat MF) and rearfoot (RF). Kinetic and EMG data were analyzed using the dominant leg. EMG signals were first modified with a 20–400 Hz band-pass filter and then a full wave rectifier to denoise. The root mean square (RMS) with 50 windows was used to analyze muscle activation. The EMG phases were defined according to previous research [10,11], and these were: pre-activation (PRE), background activity (BGA), short latency stretch-reflex component (M1) and long latency stretch-reflex component (M2).

### 2.5. Statistical Analysis

We used SPSS 21 for Windows (IBM SPSS Statistics 21.0; Somers, New York, NY, USA) to analyze the variables. The data were presented as the means and standard deviations. A 2 × 3 repeated measures ANOVA was used to compare the differences of one- and two-leg landing and footwear types during jump rope. Levene’s test was used to test data homogeneity, and Shapiro–Wilk tests were run to evaluate the normal distribution. Bonferroni post hoc analysis was then conducted when the level of significance was met. The level of significance was set at *p* < 0.05.

## 3. Results

The results are shown in Table 1, Table 2 and Table 3. Table 1 shows the results from the force platform, and the GRF, vGRF, contact time and flight time were significantly different in the two-leg landing (all *p* = 0.001).

Table 2 shows the results for plantar pressure: the GRF, GRF (BW) and Lat MF pressure were significantly greater in the one-leg landing (*p* = 0.018, 0.013 and 0.027); the pressure of the Lat MF and H area were significantly greater in the volleyball shoe (*p* = 0.025, 0.031); the pressure of the Mid FF and Lat FF area were significantly greater in the jumping shoe (*p* = 0.005, 0.042).

The EMG results of the lower extremities show that there was no significant difference found between footwear or landing conditions (Table 3).

## 4. Discussion

The purpose of this study was to examine the influence of footwear type on the lower extremity biomechanics and plantar pressure during one- and two-leg jump rope activities. Greater differences were found with different landing conditions from the force platform results. The plantar pressure results showed that the volleyball shoe demonstrates a greater pressure of the Lat MF and H areas, and the jumping shoe demonstrates a greater pressure of the Mid FF and Lat FF areas. Additionally, no difference was found in terms of lower extremity muscle-activation.

The GRF results in this research were twofold and were collected using force platform and Pedar insole. However, the GRF results of the two pieces of equipment were opposing, which was consistent with the previous research [21]; the difference was due to the different equipment calculation principles. The GRF of the force platform was calculating the whole-body effect [23], but the Pedar insole was calculating the contact areas of the sensors, which was affected by the sensor density and distribution of the foot contact areas [24]. The present results indicate that the stress of a two-leg jump was smaller than a one-leg jump, which may benefit people who use a jump rope for long-term exercise [17]. Nevertheless, no difference was found between the footwear types.

Smaller plantar pressure of Lat MF was shown in the two-leg jump rope. The pressure of the Lat MF, H, Mid FF and Lat FF areas of the running shoe were smaller than that of the volleyball shoe and jumping shoe. The H area was representing the hallux area, which was the main peak pressure area during jump rope training [20]. The current results show that the stress of the H area in the running shoe and jumping shoe was better than the volleyball shoe. The Lat MF and Lat FF were on the lateral side of the plantar which may be influenced by individual differences, such as technique [15], ankle stability [25] and leg muscle strength [21], during jump rope training. In lateral plantar pressure areas, the opposite result was found with different footwear types, that the largest Lat MF pressure and the smallest pressure in the Lat FF area were both found in the volleyball shoe and jumping shoe. If Lat MF and Lat FF were complementary, when they were regarded as separate, then the jumping shoe had a smaller pressure during jump rope training. Certainly, this needs more research to be proven. The running shoe showed the largest pressure in the midfoot (Mid FF area), which was different to previous research which may be due to the difference in footwear construction such as midsole hardness, midsole thickness or torsional stiffness. However, the experiment did not compare the construction detail of the footwear, because we mainly selected the footwear because they were easy and convenient for people to choose.

There was no significant difference found in lower extremity muscle activation in the current research. In addition, previous research indicated that the ankle was the primary implement joint during jump rope training [15,21], which means the muscle activations of TA and GA could have a greater impact during jump rope training. Although the inter-participant discrepancies in response to footwear type were evident during the experiment, there was a low inter-individual variability in muscle activation in jump rope training in all muscles. Considering the different footwear types and the different foot structures, each person may have responded differently to the shoes. This may be why acute footwear studies seem to result in no significant effects on foot and leg muscle activation, which also explains why there was no difference in EMG amplitudes between the different footwear conditions, and why there was relatively high interindividual variability in the muscle activation during jump rope training [26]. Therefore, to better understand the effects of footwear type for lower extremity biomechanical mechanisms during jump rope training, long-term studies are needed.

However, some limitations in this research should be discussed. The individual technique differences existed despite there being a basic standard for evaluating jump rope ability, which caused the standard deviation of EMG offset to be relatively high. In addition, a small sample size has a larger uncertainty, which could potentially result in type II errors, also a mix of genders could have influenced the results. Furthermore, further research could detail the influence of footwear components on jump rope training such as insole hardness or bending stiffness, which might have more benefits for understanding jump rope biomechanics and also give a standard for people to select suitable footwear.

## 5. Conclusions

The current study investigated the influence of footwear types and landing conditions on plantar pressure and lower extremity biomechanics during jump rope training. The present result showed that the main force area during jump rope training was in the forefoot area. The jumping shoe has a lower plantar pressure on the hallux, which may be beneficial for decreasing the injury risk of prolonged jump-rope training. The different footwear structure designs may influence the plantar pressure in the arch area. In summary, the running shoe and jumping shoe showed lower plantar pressure than the volleyball shoe during the experiment, which means they might be better for people who exercise. However, the running shoe is recommended for people when both jumping and running are required.

## Figures and Tables

**Figure 1 ijerph-20-01731-f001:**
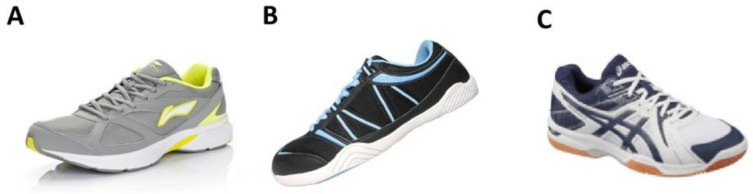
Test footwear (**A**) running shoe; (**B**) jumping shoe; (**C**) volleyball shoe.

**Table 1 ijerph-20-01731-t001:** The descriptive statistics (Mean ± SD) and statistical result of force platform variables during jump rope task.

	Running Shoe		Jumping Shoe		Volleyball Shoe		*p*-Values		
	One-Leg Jump	Two-Leg Jump	One-Leg Jump	Two-Leg Jump	One-Leg Jump	Two-Leg Jump	Interaction	Footwear	Condition
GRF (N) ^c^	1856.99 ± 334.57	2560.78 ± 341.41	1905.79 ± 288.56	2564.83 ± 346.44	1875.84 ± 270.38	2604.82 ± 296.66	*p* = 0.495	*p* = 0.432	*p* = 0.001
Peak vGRF (N) ^c^	1949.94 ± 333.03	2615.04 ± 331.77	2014.84 ± 264.95	2601.68 ± 315.96	1990.08 ± 289.94	2656.34 ± 286.45	*p* = 0.398	*p* = 0.473	*p* = 0.001
GRF (BW) ^c^	3.23 ± 0.59	4.34 ± 0.67	3.333 ± 0.51	4.46 ± 0.58	3.263 ± 0.47	4.53 ± 0.50	*p* = 0.522	*p* = 0.074	*p* = 0.001
Peak vGRF (BW) ^c^	3.39 ± 0.59	4.42 ± 0.63	3.53 ± 0.48	4.52 ± 0.53	3.46 ± 0.50	4.62 ± 0.492	*p* = 0.572	*p* = 0.078	*p* = 0.001
Contact time (ms) ^a,c^	247.90 ±52.84	181.42 ± 24.44	249.58 ± 27.91	189.60 ± 22.71	247.22 ± 25.61	182.38 ± 27.52	*p* = 0.019	*p* = 0.506	*p* = 0.001
Flight time (ms) ^c^	165.52 ± 39.31	237.26 ± 62.20	168.10 ± 42.96	223.30 ± 51.83	162.2 ± 43.11	225.52 ± 51.90	*p* = 0.254	*p* = 0.079	*p* = 0.001

^a^ means significant interaction of footwear and landing condition. ^c^ means significant difference between one- and two-leg landing during jump rope.

**Table 2 ijerph-20-01731-t002:** The descriptive statistics (Mean ± SD) and statistical result of plantar pressure variables during jump rope task.

	Running Shoe		Jumping Shoe		Volleyball Shoe		*p*-Values		
	One-Leg Jump	Two-Leg Jump	One-Leg Jump	Two-Leg Jump	One-Leg Jump	Two-Leg Jump	Interaction	Footwear	Condition
Peak force (N) ^c^	918.98 ± 195.54	798.21 ± 268.75	1072.14 ± 245.31	815.02 ± 330.64	966.66 ± 214.39	728.19 ± 242.11	*p* = 0.326	*p* = 0.168	*p* = 0.018
Peak force (BW) ^c^	1.51 ± 0.24	1.32 ± 0.41	1.76 ± 0.29	1.30 ± 0.51	1.59 ± 0.26	1.16 ± 0.32	*p* = 0.278	*p* = 0.178	*p* = 0.013
Peak pressure (kpa)	464.55 ± 185.45	475.85 ± 157.48	477.39 ± 161.58	487.58 ± 145.25	437.3 ± 197.50	398.39 ± 175.97	*p* = 0.805	*p* = 0.388	*p* = 0.861
Average pressure (kpa)	107.48 ± 41.54	113.60 ± 48.46	126.50 ± 61.18	116.69 ± 40.41	96.16 ± 31.07	105.36 ± 42.77	*p* = 0.679	*p* = 0.096	*p* = 0.857
RF (kpa)	24.35 ± 39.38	0 ± 0	39.18 ± 51.23	0.10 ± 0.30	12.93 ± 18.03	0.09 ± 0.29	*p* = 0.119	*p* = 0.120	*p* = 0.067
Lat MF (kpa) ^b,c^	106.99 ± 73.59	48.42 ± 51.68	92.57 ± 61.24	37.82 ± 40.25	134.58 ± 69.70	74.40 ± 66.38	*p* = 0.857	*p* = 0.025	*p* = 0.027
Med MF (kpa)	45.92 ± 41.57	16.12 ± 17.54	35.24 ± 22.83	14.96 ± 19.14	46.21 ± 32.52	19.52 ± 28.57	*p* = 0.446	*p* = 0.160	*p* = 0.067
Med FF (kpa)	228.46 ± 164.65	293.78 ± 151.93	280.17 ± 173.24	350.64 ± 209.67	205.14 ± 132.97	274.47 ± 132.92	*p* = 0.836	*p* = 0.052	*p* = 0.462
Mid FF (kpa) ^b^	313.94 ± 201.44	411.71 ± 255.30	377.07 ± 196.78	452.85 ± 225.48	246.40 ± 110.97	357.85 ± 146.02	*p* = 0.425	*p* = 0.005	*p* = 0.331
Lat FF (kpa) ^b^	126.24 ± 64.72	136.62 ± 86.20	152.10 ± 107.04	151.97 ± 106.63	105.99 ± 60.74	111.67 ± 73.86	*p* = 0.408	*p* = 0.042	*p* = 0.839
H (kpa) ^b^	102.53 ± 58.12	159.32 ± 80.69	96.86 ± 56.52	144.68 ± 83.42	134.09 ± 70.80	189.55 ± 54.51	*p* = 0.997	*p* = 0.031	*p* = 0.267
LT (kpa)	103.62 ± 89.05	189.86 ± 168.91	65.13 ± 35.86	129.96 ± 118.42	113.44 ± 148.62	160.8 ± 112.79	*p* = 0.943	*p* = 0.564	*p* = 0.216
P × T (kpa∙s)	562.46 ± 641.62	621.99 ± 526.39	518.58 ± 424.68	570.86 ± 583.42	454.61 ± 558.98	454.82 ± 530.79	*p* = 0.628	*p* = 0.854	*p* = 0.073
F × T (N∙s)	814.58 ± 569.73	869.54 ± 530.16	888.46 ± 595.06	855.99 ± 812.11	720.65 ± 532.76	667.94 ± 525.94	*p* = 0.823	*p* = 0.494	*p* = 0.337

^b^ means significant difference between footwear. ^c^ means significant difference between one- and two-leg landing during jump rope. H = hallux; LT = lesser toes; Med FF = medial forefoot; Mid FF = middle forefoot; LatFF = lateral forefoot; Med MF = medial midfoot; Lat MF = lateral midfoot; RF = rearfoot; P = pressure; T = time; F = force.

**Table 3 ijerph-20-01731-t003:** The descriptive statistics (Mean ± SD) and statistical result of EMG variables during jump rope task.

Unit: %MVC		Running Shoe		Jumping Shoe		Volleyball Shoe		*p*-Values		
		One-Leg Jump	Two-Leg Jump	One-Leg Jump	Two-Leg Jump	One-Leg Jump	Two-Leg Jump	Interaction	Footwear	Condition
Pre	VM	43.08 ± 12.38	72.52 ± 29.85	61.61 ± 21.82	38.71 ± 11.14	29.79 ± 18.27	237.74 ± 53.63	*p* = 0.199	*p* = 0.394	*p* = 0.188
	BF	35.41 ± 10.31	20.31 ± 16.60	26.60 ± 18.83	25.03 ± 15.16	28.12 ± 15.87	61.84 ± 71.23	*p* = 0.119	*p* = 0.231	*p* = 0.236
	TA	30.77 ± 14.76	40.18 ± 13.00	63.62 ± 22.48	47.82 ± 13.72	34.44 ± 10.26	32.07 ± 29.51	*p* = 0.624	*p* = 0.410	*p* = 0.552
	GA ^a^	89.72 ± 27.64	55.26 ± 14.39	75.69 ± 16.15	71.66 ± 24.51	92.34 ± 37.54	166.92 ± 166.08	*p* = 0.018	*p* = 0.231	*p* = 0.428
BGA	VM	58.75 ± 16.54	58.71 ± 13.74	60.38 ± 13.92	40.82 ± 11.58	31.54 ± 19.81	190.43 ± 41.48	*p* = 0.241	*p* = 0.527	*p* = 0.392
	BF	37.13 ± 15.79	32.78 ± 18.42	30.19 ± 13.85	26.06 ± 18.50	27.06 ± 17.80	46.61 ± 13.04	*p* = 0.148	*p* = 0.601	*p* = 0.643
	TA	36.41 ± 18.92	26.01 ± 26.85	33.15 ± 15.55	39.87 ± 11.71	27.47 ± 12.01	34.22 ± 18.86	*p* = 0.119	*p* = 0.741	*p* = 0.803
	GA	104.51 ± 23.49	68.48 ± 25.71	92.92 ± 34.65	89.23 ± 31.74	93.25 ± 31.49	94.37 ± 31.54	*p* = 0.580	*p* = 0.967	*p* = 0.574
M1	VM	116.84 ± 36.72	112.46 ± 34.34	89.71 ± 39.66	81.29 ± 35.56	24.19 ± 16.09	267.37 ± 50.86	*p* = 0.163	*p* = 0.299	*p* = 0.202
	BF	34.07 ± 16.60	32.28 ± 17.73	25.67 ± 9.59	37.26 ± 10.63	24.53 ± 16.39	45.22 ± 19.36	*p* = 0.237	*p* = 0.950	*p* = 0.228
	TA	42.93 ± 11.64	39.88 ± 11.62	26.56 ± 9.21	36.42 ± 11.15	22.33 ± 18.80	44.93 ± 11.50	*p* = 0.098	*p* = 0.658	*p* = 0.257
	GA	113.84 ± 21.28	79.51 ± 22.26	105.19 ± 35.93	93.07 ± 34.55	123.60 ± 32.23	93.96 ± 28.22	*p* = 0.890	*p* = 0.926	*p* = 0.370
M2	VM	116.77 ± 31.34	103.64 ± 35.26	80.65 ± 26.86	73.22 ± 22.65	26.13 ± 18.76	191.53 ±43.33	*p* = 0.260	*p* = 0.376	*p* = 0.261
	BF	37.60 ± 16.96	27.92 ± 12.97	30.17 ± 17.97	33.48 ± 11.88	26.90 ± 14.54	33.16 ± 16.09	*p* = 0.342	*p* = 0.942	*p* = 0.995
	TA	44.99 ± 12.01	39.32 ± 11.47	29.87 ± 16.82	40.47 ± 17.01	22.93 ± 17.018	48.36 ± 17.01	*p* = 0.082	*p* = 0.792	*p* = 0.294
	GA	110.76 ± 22.51	84.50 ± 26.30	124.37 ± 36.89	87.02 ± 25.53	136.64 ± 35.45	69.96 ± 22.32	*p* = 0.702	*p* = 0.968	*p* = 0.126

^a^ means significant interaction of footwear and landing condition.

## Data Availability

Data are contained within the article.

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
