# Peer review of "Effects of Footwear Selection on Plantar Pressure and Neuromuscular Characteristics during Jump Rope Training"

_ijerph, 2023, doi:10.3390/ijerph20031731_

Round 1

Reviewer 1 Report

Introduction:

Please re-read for general grammar and sentence structure edits

Line 38 - 39: add 'are' in between 'opportunities' and 'always' so the sentence reads "...training opportunities are always limited...".

Line 42: add 'is' in between 'that' and 'also' so the sentence reads '...that is also suitable for...'

Methods:

Line 68 - 69: please add participant demographic information (i.e., mean and +/- standard deviation of age, weight, height). 

Line 72: please add if there were any requirements of participant activity level (i.e., any requirements of recreational activity multiple days a week, any activity requiring jumping, etc.)

Line 88: please indicated if shoe condition were counterbalanced or randomized.

Line 89: please indicate how leg dominance was defined

Line 93: please indicate if the one and two leg jump conditions were counterbalanced or randomized

Line 93: please indicate how much rest participants received between the one and two leg jump conditions

Line 97: please indicate how much rest was given between each shoe condition

Discussion:

Line 133 - 135: please define the acronyms Lat MF, H, Mid FF, and Lat FF. These acronyms are also used in Table 2 and need to be defined

Line 178: please add that sex is a limiting factor for the study. As there were a mix of 5 males and 5 females, sex must be addressed as a limitation that could have influenced the results

Author Response

Introduction:

Please re-read for general grammar and sentence structure edits

Response: Thanks for your comments. The manuscript has been edited by MDPI English editing services.   

Line 38 - 39: add 'are' in between 'opportunities' and 'always' so the sentence reads "...training opportunities are always limited...".

Response: Thanks for your comments. It’s have been revised. Please refer to line 38.

Line 42: add 'is' in between 'that' and 'also' so the sentence reads '...that is also suitable for...'

Response: Thanks for your comments. It’s have been revised. Please refer to line 42.

Methods:

Line 68 - 69: please add participant demographic information (i.e., mean and +/- standard deviation of age, weight, height).

Response: Thanks for your comments. These have been revised. Please refer to line 70-71.

Line 72: please add if there were any requirements of participant activity level (i.e., any requirements of recreational activity multiple days a week, any activity requiring jumping, etc.)

Response: Thanks for your comments. These have been revised. Please refer to line 71-72.

Line 88: please indicated if shoe condition were counterbalanced or randomized.

Response: Thanks for your comments. These have been revised. Please refer to line 91.

Line 89: please indicate how leg dominance was defined

Response: Thanks for your comments. These have been revised. Please refer to line 92.

Line 93: please indicate if the one and two leg jump conditions were counterbalanced or randomized

Response: Thanks for your comments. These have been revised. Please refer to lines 95-98.

Line 93: please indicate how much rest participants received between the one and two leg jump conditions

Line 97: please indicate how much rest was given between each shoe condition

Response: Thanks for your comments. These have been revised. Please refer to lines 100-101.

Discussion:

Line 133 - 135: please define the acronyms Lat MF, H, Mid FF, and Lat FF. These acronyms are also used in Table 2 and need to be defined

Response: Thanks for your comments. The areas of plantar was divided into 8 regions which were hallux (H), lesser toes (LT), medial forefoot (Med FF), middle forefoot (Mid FF), lateral forefoot (LatFF), medial midfoot (Med MF), lateral midfoot (Lat MF), and rearfoot (RF). It has been replenished in lines 108-110.

Line 178: please add that sex is a limiting factor for the study. As there were a mix of 5 males and 5 females, sex must be addressed as a limitation that could have influenced the results

Response: Thanks for your comments. These have been revised. Please refer to line 189.

Reviewer 2 Report

Dear authors, the research:                                                                                                                                                                                                                    

“Effects of Footwear Selection on Plantar Pressure and Neuromuscular Characteristics During Jump Rope Training”  it is  very interesting; We would like to some considerations that we consider:

Material and method

There is no any information about the allocation or origin of the sample is not described in the text, nor is the type of sample that was carried out.

The sample size is too small, consider that the investigation consists of a pilot or preliminary research. It is poorly described the methodology

Author Response

“Effects of Footwear Selection on Plantar Pressure and Neuromuscular Characteristics During Jump Rope Training”  it is  very interesting; We would like to some considerations that we consider:

Response: We appreciate your kind advice. The manuscript has been revised

Material and method

There is no any information about the allocation or origin of the sample is not described in the text, nor is the type of sample that was carried out.

Response: Thanks for your comments. These have been revised. Please refer to lines 70-71.

The sample size is too small, consider that the investigation consists of a pilot or preliminary research. It is poorly described the methodology.

Response: Thanks for your comments. The sample size was a limitation in this study. Please refer to lines 187-188. However, we have used G*Power software to explained the data reliability.  Please refer to lines 72-74.  

The part methodology has revised.

Round 2

Reviewer 1 Report

Than you to the authors for addressing my comments. I have no further comments at this time.

Author Response

Than you to the authors for addressing my comments. I have no further comments at this time.

Response: We appreciate your kind advice.

Reviewer 2 Report

Dear authors, I considerate the manuscript has been improved, any way, maybe it is pertinent to include it is a pilot study in the title

Author Response

Dear authors, I considerate the manuscript has been improved, any way, maybe it is pertinent to include it is a pilot study in the title

Response: We appreciate your kind advice. The title of this study has been revised.